# The Role of Peroxiredoxins in the Regulation of Sepsis

**DOI:** 10.3390/antiox11010126

**Published:** 2022-01-06

**Authors:** Toshihiko Aki, Kana Unuma, Koichi Uemura

**Affiliations:** Department of Forensic Medicine, Graduate School of Medical and Dental Sciences, Tokyo Medical and Dental University, Tokyo 113-8519, Japan; unumlegm@tmd.ac.jp (K.U.); kuemura.legm@tmd.ac.jp (K.U.)

**Keywords:** peroxiredoxins, sepsis, inflammasome, pyroptosis, inflammation

## Abstract

Oxidative stress, a result of a disturbance in redox homeostasis, is considered to be one of the main aggravating events in the pathogenesis of immune disorders. Peroxiredoxins (Prdxs) are an enzyme family that catalyzes the reduction of peroxides, including hydrogen peroxide, lipid peroxides, and nitrogen peroxides. Although the maintenance of cellular redox homeostasis through Prdxs is essential for surviving in adverse environments, Prdxs also participate in the regulation of cellular signal transduction by modulating the activities of a panel of molecules involved in the signal transduction process. Although Prdxs were discovered as intracellular anti-oxidative enzymes, recent research has revealed that Prdxs also play important roles in the extracellular milieu. Indeed, Prdxs have been shown to have the capacity to activate immune cells through ligation with innate immune receptors such as toll-like receptors (TLRs). In this review, we will summarize the intracellular as well as extracellular roles of Prdxs for and against the pathogenesis of inflammatory disorders including sepsis, hemorrhagic shock, and drug-induced liver injury.

## 1. Introduction

### 1.1. Sepsis

Sepsis is a multi-organ disease, observed mainly in ICU patients, which is associated with high mortality rates that reach ~25% for cases of sepsis and ~45% for cases of septic shock [1,2]. A systemic but disordered immune reaction leads not only to fever and tiredness, but also to the systemic activation of coagulation and subsequent insufficient oxygenation of multiple organs [3]. Therefore, sepsis is associated with disseminated intravascular coagulation (DIC), multi-organ failure (MOF), and shock [3]. During infection by pathogens, host recognition of molecular patterns of invading microorganisms (pathogen-associated molecular patterns, PAMPs), as well as damage-associated molecular patterns (DAMPs) derived from endogenous cells, leads to innate and adaptive immune responses that must abate for the recovery of healthy status, but that persist for sustained periods during sepsis (Figure 1). The activation of inflammasomes and subsequent pyroptosis play critical roles in the etiology of sepsis. Another characteristic feature in the pathophysiology of sepsis is mitochondrial dysfunction [4]. In accordance with the lack of sufficient oxygenation, mitochondrial dysfunction and subsequent oxidative stress have been implicated in the etiology of sepsis; mitochondrial dysfunction should be involved not only in redox imbalance and oxidative stress, but also in the cellular energy deficiency observed during sepsis [4,5].

The definition of sepsis was formerly determined by the American College of Chest Physicians and the Society of Critical Care Medicinein 1991, as the presence of both infection and subsequent systemic inflammatory response syndromes (SIRSs), the latter defined by the presence of symptoms such as fever, tachycardia, tachypnoea, leukopaenia, or others [6]. However, these symptoms are observed in both septic and non-septic critically ill patients. Therefore, sepsis has now been re-defined as a dysregulated immune response to infection associated with life-threating organ failure (Sepsis-3 criteria) [7,8,9,10]. The sepsis bundle has also been updated; please refer the references [7,11] for detailed information about the sepsis bundle.

### 1.2. Inflammasomes and Pyroptosis

Upon infection by invasive pathogens, inflammatory cells are recruited to the sites of infection. In the early stages of infection, immune cells such as neutrophils and macrophages, which are implicated in the innate immune response, are activated, after which cytokines and chemokines are produced to further activate the adaptive immune responses. Inflammasomes play a central role in the activation of the innate immune system [12,13,14]. During infection by pathogens, PAMPs are recognized by their cognate pattern-recognition receptors (PRRs) [15,16]. For example, lipopolysaccharides (LPS), a cell wall component of gram-negative bacteria, are recognized by toll-like receptor 4 (TLR4) [17,18,19], while flagellin, the major protein in bacterial flagella, is recognized by TLR5 [20,21]. DAMPs, which are molecules released from damaged cells and have molecular patterns similar to PAMPs, can also elicit immune responses by ligating PPRs [22,23,24].

One of the most important responses of inflammatory cells during the innate immune response is the release of pro-inflammatory cytokines, which is executed through the activation of the inflammatory caspases, caspase-1 and -11 (the human equivalents of murine caspase-11 are caspase-4 and -5) [25,26]. Inflammasomes, multiprotein complexes activated in response to the ligation of PRR by PAMPs and DAMPs, are involved in the activation of caspase-1 and -11 (Figure 2) [12,13,14]. Caspase-1 and -11 are required for the processing as well as maturation of proinflammatory cytokines such as IL-1β, IL-12, and IL-18. Inflammasomes have been shown to play critical roles in the etiology of sepsis [12]. There are two pathways that can lead to inflammasome activation, the canonical and non-canonical pathways [27]. The canonical pathway, which is triggered by pathogen infection and the resultant formation of inflammasomes, is responsible for the activation of caspase-1 [12]. Canonical inflammasome activation comprises two steps. The priming step (also called signal 1) consists of PRR activation and the subsequent induction of the expression of inflammasome component genes, which is mediated by transcription factors including NF-κB. The activation step (signal 2) includes inflammasome assembly as well as the cytosolic efflux of K^+^ ions, a crucial event required for inflammasome activation [28,29]. In case of the NLRP3 (NOD-like receptor family pyrin domain-containing 3) inflammasome, inflammasome components, NLRP3, ASC (apoptosis-associated speck-like protein containing a CARD), and caspase-1 are assembled to activate caspase-1 [12]. In contrast to the canonical pathway, which is dependent on TLR4, the non-canonical pathway of inflammasome activation is triggered by the direct binding of intracellular LPS to caspase-11, which leads to the activation of this caspase in a TLR4-independent manner [30,31].

The release of IL-1β, IL-18, and other small molecules is mediated by the gasdermin D (GSDMD)-dependent formation of plasma membrane channels [32,33,34,35] (Figure 2). Both caspase-1 and caspase-11 cleave GSDMD at the same site (Arp276 in mouse), liberating the N-terminus half of GSDMD, which forms pores in the plasma membrane through which IL-1β and IL-18 can be secreted into the extracellular milieu. In addition, these pro-inflammatory molecules can be released when the plasma membrane ruptures, resulting in necrotic cell death. Pyroptosis is a lytic form of cell death that is morphologically classified as necrosis, but which is executed in a strictly regulated manner [26,36,37,38]. Pyroptosis is the major mode of cell death in the etiology of sepsis, and is involved in the propagation of inflammation by releasing pro-inflammatory molecules from the cells [39]. It had been assumed that water influx from the extracellular environment into the intracellular space through GSDMD pores was the causative mechanism of plasma membrane rupture during pyroptosis. However, recent research has identified Ninjurin 1(NINJ1) as a key molecule in plasma membrane rupture during pyroptosis; loss of NINJ1 suppresses cell rupture [40]. Although the precise mechanism of the membrane rupture caused by NINJ1 has not been elucidated, necrosis and the subsequent rupture of the plasma membrane might be a much more highly regulated process than previously assumed [41].

### 1.3. Ferroptosis

Ferroptosis is another form of necrotic cell death implicated in immune cell death [42,43,44]. In the same manner as pyroptosis, ferroptosis is a form of regulated necrosis characterized by its dependency on ferrous iron as well as the formation of lipid peroxides (Figure 3) [43,45]. It has been shown that in healthy cells, lipid peroxides are eliminated through the action of glutathione peroxidase 4 (Gpx4), the only member of the Gpx protein family that can reduce lipid peroxides [45,46]. Kang et al. have shown that Gpx4 levels are specifically increased among the Gpx family members in peritoneal macrophages as well as peripheral blood mononuclear cells in animal models of sepsis created by the cecal ligation of puncta (CLP) [47]. Thus, there should be an increased demand for the elimination of lipid peroxides from these cells in CLP mice. The conditional knock-out of Gpx4 in myeloid cells results in an acceleration of the lethality of CLP sepsis, proving the pivotal and detrimental role of lipid peroxidation in CLP mice [47]. Interestingly, the death of myeloid cells in CLP mice is associated not only with lipid peroxidation, but also with inflammasome activation, GSDMD cleavage, and increased levels of circulating cytokines, suggesting the occurrence of pyroptosis [47]. Knock-out of Gpx4 in CLP mice leads to caspase-11/GSDMD-dependent pyroptosis in myeloid cells, confirming that lipid peroxidation is involved not only in ferroptosis, but also in pyroptosis [47].

### 1.4. Release Mechanism of DAMPs from Cells

PAMPs are derived from outside pathogens that invade the host whereas DAMPs are self-derived. DAMPs are involved in the pathogenesis of sepsis by forming a positive feedback loop for the propagation of inflammation; the activation of inflammasomes through PAMPs also results in the release of DAMPs, which circulate throughout the body and further propagate inflammation [48]. Even without pathogenic infection, DAMPs can be released from damaged cells to mediate inflammation. This sterile inflammation is considered to be involved in the immune reactions that take place during trauma, ischemia-reperfusion injury, and drug-induced injuries such as DILI (drug-induced liver injury) [23]. GSDMD pores in the plasma membrane can accommodate molecules smaller than approximately 20 kDa [35]. Thus, even in the absence of cell death, low molecular weight DAMPs can be released from cells when inflammasome activation and the resultant GSDMD pore formation occur (Figure 4). When inflammasome activation reaches the stage of cell death, essentially all the molecules within the cell can be released into the extracellular milieu. Cell death such as that caused by pyroptosis and ferroptosis is also important in inflammation, as intracellular contents including cytokines (e.g., TNFα, IL-1β) and DAMPs (e.g., HMGB1 ATP) are released into the extracellular milieu through rupture of the plasma membrane [23].

There are other mechanisms by which cytokines and DAMPs can be released from cells [49]. In addition to the release of DAMPs from ruptured cells or through pores formed in the plasma membrane, exocytosis is also involved in the release of DAMPs (Figure 4). Exocytosis can take place through a variety of routes [50]. In addition to ordinary exocytosis during which secretory vesicles are released from cells, the contents of lysosomes can be released through lysosomal exocytosis [50]. Among the many routes toward exocytosis, the multivesicular body (MVB)-mediated route should play an important role in the regulation of sepsis. MVB-mediated exocytosis is involved in the release of exosomes [51,52], and it has been demonstrated that exosomes are crucially involved in the regulation of sepsis [53,54,55,56]. Exosomes have been shown to deliver not only micro RNAs but also DAMPs [57].

## 2. Prdxs

Peroxiredoxins (Prdxs) comprise a family of thiol-specific peroxidases that catalyze the reduction of not only hydrogen peroxide but also alkyl peroxides as well as peroxynitrites [54,58,59]. There are six Prdxs found in mammals: Prdx1–6. These six Prdxs are classified into three groups based on their structures and the mechanism of catalysis (Figure 5), although all Prdxs form dimers [60]. The 2-Cys Prdxs, Prdxs1–4, contain two Cys residues that are involved in the reduction of peroxides [60]. In the 2-Cys Prdxs, there are two distinct types of Cys, peroxidatic and resolving Cys (C_P_ and C_R_, respectively). Because the 2-Cys Prdxs form dimers in which two Prdx monomers are arranged in an antiparallel fashion, the C_P_ in one Prdx is arranged face-to-face with the C_R_ in the other Prdx [61]. Due to the structural characteristic of the C_P_ being surrounded by basic amino acids such as His and Arg, peroxides can gain easy access to the C_P_ in Prdxs [61]. The binding of hydroperoxide to C_P_ in one Prdx molecule results in the formation of an unstable intermediate, cysteine sulfenic acid (C_P_-SOH), which subsequently forms an intermolecular disulfide bond with the C_R_ in the other Prdx. This disulfide-bonded and oxidized dimer is returned to the non-disulfide-bonded and reduced dimer through the action of thioredoxin (Trx). Although Prdx5 contains both C_P_ and C_R_, its mechanism for the reduction of peroxides is distinct from that of the 2-Cys Prdxs, and, therefore, Prdx5 is classified as an atypical 2-Cys Prdx [62]. In contrast to the typical 2-Cys Prdxs, Prdx5 forms dimers in which the two monomers are not arranged in an anti-parallel manner [63,64]. Prdx5 forms an intramolecular disulfide bond between the C_P_ and C_R,_ within one Prdx molecule and is returned to its reduced form though the action of Trx [65]. Prdx6 lacks the C_R_ and is thus classified as a 1-Cys Prdx [66,67]. In contrast to the dependence of the reduction of Prdx1–5 on Trx, the disulfide bond in the oxidized form of Prdx6 is reduced by glutathione S-transferase π isoform (GSTπ) [68].

It should be noted that cysteine sulfenic acid (C_P_-SOH) is sometimes further oxidized by peroxides resulting in the formation of C_P_-SO_2_H within the Prdxs [69]. Although several roles for this hyperoxidation of Prdx have been suggested, such as in protecting cells from oxidative stress, functioning as a chaperone, and regulating specific signal transductions, there remains little explanation for the significance of the hyperoxidation of Prdx [70]. Because hyperoxidized Prdx is reduced exclusively by sulfiredoxin (Srx) and Srx seems to have evolved exclusively for the reduction of hyperoxidized Prdx, there should be substantial significance to hyperoxidized Prdx and its reduction by Srx in the maintenance of cellular homeostasis as well as the whole organism under oxidative stress [71,72]. Although all Prdxs form dimers irrespective of inter- or intra-molecular disulfide bond formation, typical 2-Cys Prdxs oligomerize further to form decamers or even larger oligomers [73]. Although the functional significance of the dimer–decamer transition remains unknown, it has been suggested that the transition of Prdx1 from decamers to dimers results in the loss of its chaperone activity [74]. 

Recent research has revealed that when Prdxs are released from cells, they can act as DAMPs [75,76]. Among PRRs, TLRs are the main receptors responsible for the action of Prdxs as DAMPs [75,76].

### 2.1. Prdx1

Prdx1 was the first Prdx shown to act as a DAMP when released into the extracellular space [75] (Figure 6). Riddell et al. reported that Prdx1 when added extracellularly can stimulate the TLR4-dependent secretion of TNFα as well as IL-6 from macrophages and dendritic cells [75]. They demonstrated that both recombinant Prdx1 and the supernatant from Prdx1-secreting tumor cells could cause these cells to secrete TNFα and IL-6 [75]. Supernatant from tumor cells in which Prdx1 was knocked down could not induce this secretion [75]. This ability of extracellular Prdx1 to elicit TNFα and IL-6 secretion from immune cells was also observed in vivo; they demonstrated that the injection of recombinant Prdx1 into mice resulted in an elevation of systemic IL-6 levels [75]. This IL-6 increase could not be replicated in TLR4 knock-out (KO) mice, confirming the essential role of TLR4 [75]. Interestingly, they observed that the Prdx1C52S mutant, which lacks peroxidase activity but forms decamers, could increase Il-6 levels to the same extent as wild type Prdx1, while the Prdx1C83S mutant, which retains intact peroxidase activity but cannot form decamers, increased IL-6 to a lower level than wild type Prdx1 [75]. Thus, it is not its enzymatic activity, but its structural characteristics that seem to be important for Prdx1 to act as a DAMP. It should be noted that the authors, taking into consideration a report demonstrating that HMGB1 binding to TLR9 is mediated by not HMGB1 itself but DNA associated with HMGB1 [77], suggested the possibility that Prdx1 itself does not interact with TLR4 directly [75]. The propagation of inflammation by secreted Prdx1 has also been reported not only in animal models of septic shock [78], but also in drug-induced acute liver injury (DILI) [79], confirming the role of extracellular Prdx1 as a DAMP.

On the other hand, Prdx1 can bind to caspase-1 to suppress inflammasome activation within immune cells such as macrophages [80]. Liu et al. conducted a screening of drugs possessing anti-inflammatory properties, and identified AI-44, a curcumin analogue, as having the ability to bind to Prdx1 [80]. The binding of AI-44 to Prdx1 results in the promotion of the interaction between Prdx1 and caspase-1, which also leads to the dissociation of the NLRP3–ASC–caspase-1 complex, inactivation of the NLRP3 inflammasome, and negative regulation of inflammation [80]. Indeed, AI-44 mitigates LPS-induced endotoxemia in mice, confirming the negative regulation of inflammasomes by Prdx1 [80]. Therefore, the roles of intracellular and extracellular Prdx1 on inflammasomes seem to be totally different. In addition, there are discrepancies between the role of Prdx1 in whole body homeostasis during sepsis. Although one report demonstrates that Prdx1 deficiency mitigates lethality in an LPS-induced animal model of septic shock [78], another report indicates an acceleration of death due to LPS-model sepsis in Prdx1-KO animals [81]. Thus, further study is needed to elucidate the precise mechanism of Prdx1 in the etiology of sepsis.

### 2.2. Prdx2

In the same manner as Prdx1, extracellular Prdx2 can act as a DAMP. Prdx2 is the main Prdx protein in erythrocytes and is released from the cells during hemolysis. Intracerebral hemorrhage is followed by hemolysis, and all of the three most abundant erythrocyte proteins, hemoglobin, carbonic anhydolase-1, and Prdx2, are implicated in the development of brain edema; it has been demonstrated that the injection of these proteins into the brain can cause edema [82,83]. Bian et al. showed that the intracaudate injection of lysed red blood cells into rat brain increases Prdx2 levels, suggesting the diffusion of Prdx2 [83]. The injection of Prdx2 induces several brain disorders including blood brain barrier breakdown, neutrophil infiltration, and neuronal cell death, all of which are associated with hemorrhagic brain injury [83]. Brain swelling as well as neutrophil infiltration caused by the injection of lysed red blood cells can be suppressed by the co-administration of the Prdx2 inhibitor conoidin A [84], suggesting a role of Prdx2 in these disorders during hemorrhagic brain injury [83]. Although the precise mechanism for the role of extracellular Prdx2 in the pathogenesis of brain hemorrhage has not been elucidated, the activation of TLR4 by extracellular Prdx1 has also been reported to be involved in the neuroinflammation that occurs during intracerebral hemorrhagic injury [85]. Given the structural similarity between Prdx1 and Prdx2, it might be reasonable to speculate that, in the same manner as Prdx1, Prdx2 can also act as a TLR4 ligand in the brain. However, the precise mechanism for the action of Prdx2 in the disorders that occur during brain hemorrhage needs to be clarified in the future. 

During an attempt to identify proteins that are oxidized through thiol glutathionation and released from macrophages in response to LPS stimulation, Salzano et al. identified Prdx2 as a protein that is released from cells after glutathionation and subsequently form disulfide-linked dimers [86] (Figure 6). Prdx2 release has also been observed in cells other than macrophages, such as in HEK293 human embryonic kidney cells [86]. The release of Prdx2 is associated with its substrate Trx1, suggesting that the oxidoreduction of the Prdx2-Trx1 system might operate even in extracellular spaces [86]. Thus, the authors discussed the possible role of extracellular Prdx2-Trx1 in the oxidoreduction of membrane receptors to activate inflammatory cells [86]. Indeed, many membrane receptors can be activated after disulfide-linked dimerization (oxidation), and Trx can oxidize their substrates depending on the redox environment in which they operate [86]. Although the extracellular role of Prdx2-Trx1 as an oxidoreductase has not been confirmed, this hypothesis suggests another possibility that Prdx2 acts as a DAMP not only through the ligation of PRRs, but also through the redox regulation of cytokine receptors. In a subsequent study by the same group, Mullen et al. observed that both Prdx1 and Prdx2 are released from HEK293 cells as well as monocytic cells as dimers within exosomes following stimulation by LPS as well as TNFα [87]. They also observed that the mutation of either the C_P_ or C_R_ in Ptdx1 or Prdx2 prevents their secretion, confirming the pivotal role of cysteine oxidation in the secretion of Prdx1 and Prdx2 [87]. Thus, not only the structure, but also the redox-regulating activity of Prdx seems to be important for its secretion as well as pro-inflammatory activity outside cells.

### 2.3. Prdx3

Prdx3 is unique among Prdxs in that it localizes mainly to mitochondria. There are few reports showing the involvement of Prdx3 in the etiology of sepsis, although there is one report that Prdx3 mRNA can be found among the 84 mRNAs whose levels in exosomes are higher in patients suffering from sepsis than in healthy controls [56]. Although little is known about the possible involvement of Prdx3 in sepsis, there are examples of inflammasome regulation and inflammatory cell death by Prdx3. Acetaminophen (APAP) is one of the most commonly used pain relievers and fever reducers, but overdoses of APAP lead to acute liver injury due to the generation of its toxic metabolite, n-acetyl-p-benzoquinone imine (NAPQI) [88,89]. The excessive amounts of glutathione needed to eliminate toxic NAPQI results in oxidative stress in cells, which finally leads to mitochondrial dysfunction, inflammasome activation, and cell death through apoptosis as well as pyroptosis [89]. Wang, et al. studied animal models of drug-induced acute liver injury caused by the intraperitoneal administration of APAP or carbon tetrachloride, and found the processing of GSDMD into its p30 pore forming fragment, procaspase-1 into its p20 active fragment, and pro-IL-1β into its mature form, suggesting inflammasome-dependent pyroptosis in APAP-induced acute liver injury [90]. The activation of inflammasome-dependent pyroptosis by APAP was also observed in primary cultures of hepatocytes as well as Kupffer cells, indicating that both liver parenchymal cells and immune cells contribute to acute liver injury [90]. They also demonstrated that APAP-induced liver pyroptosis could be ameliorated and reduced by the siRNAs for NLRP3 and Prdx3, showing that liver injury by APAP is dependent on NLRP3 inflammasomes and negatively regulated by Prdx3 [90]. The negative regulation of NLRP3 inflammasomes by Prdx3 is cancelled by a scavenger of mitochondrial ROS (Mito-TEMPO) [90]. Prdx3 knock-out was found to increase mitochondrial ROS, indicating that Prdx3 negatively regulates NLRP3 activation through its antioxidative effects on mitochondria [90] (Figure 6). The amelioration of the pathogenesis of several inflammatory injuries, such as ischemia reperfusion injury of the intestine [91] and traumatic brain injuries [92], by Prdx3 has also been reported and demonstrated to be dependent on its preserving effects on mitochondrial function. Therefore, in contrast to the direct association and negative regulation of Prdx1 with inflammasome components, the effects of Prdx3 on inflammatory disorders seems to be mediated through its action on mitochondria.

### 2.4. Prdx4

In the same manner as Prdx3, Prdx4 has a unique feature in its localization; it is located in the ER and has a leader peptide for secretion in its amino acid sequence [93,94]. Indeed, Prdx4 levels in the serum of patients critically ill with sepsis and admitted to the ICU have been reported to be elevated as compared to control patients, and the levels of circulating Prdx4 correlate well with the severity of disease [95]. Consistent with this report, we have also observed that Prdx4 is released from hepatocytes in response to LPS stimulation, suggesting extracellular roles of Prdx4 in addition to its role as an intracellular anti-oxidative enzyme [96]. It has recently been reported that Prdx4 is involved in the regulation of inflammasomes in extracellular vesicles [97] (Figure 6). In Prdx4-KO mice, it has been observed that weight loss in response to a challenge with a sublethal dose of LPS is increased as compared with wild type littermates [97]. This increase in weight loss is associated with higher levels of cytokines as well as chemokines, such as IL-1β, TNF-α, and cxcl1, in the plasma, suggesting increased inflammatory responses in Prdx4-KO mice upon sublethal LPS stimulation [97]. In addition to these observations, the authors also showed that myeloid cell-specific Prdx4 deficiency in mice leads to the release of IL-1β from macrophages in response to LPS challenge, although no IL-1β from macrophages can be observed in wild type mice when LPS alone is administered [97]. In addition to LPS stimulation, an increase in the release of IL-1β from the macrophages of myeloid-specific Prdx4 knockout mice was observed in response to the administrations of PAMPs such as double-stranded DNA, flagellin, or ATP [97]. Thus, Prdx4 should be involved in the negative regulation of inflammasomes. Indeed, the authors demonstrated that Prdx4 inhibits caspase-1 activation through direct interaction: disulfide bond exchange between the cysteine 397 in caspase-1 and some cysteine residues in Prdx4 results in the sequestration of caspase-1 into high molecular weight (HMW) Prdx4 decamers [97]. The cysteine 362 in caspase-1 also seems to be involved in this sequestration into HMW Prdx4 [97]. Thus, Ptdx4 is secreted within exosomes together with inflammasomes, and is involved in the negative regulation of inflammasomes.

### 2.5. Prdx5

Shichida et al. reported that Prdxs are released from necrotic cells during ischemic brain injury, and are involved in post-ischemic inflammation by activating macrophages [76]. They screened brain lysates for DAMPs that can induce IL-23 in bone marrow-derived cells (BMDCs), and identified Prdxs as the DAMPs responsible for the induction of IL-23 in BMDCs [76]. This IL-23-inducing ability is mediated, at least in part, by TLR2 and TLR4 [76]. Among Prdxs, Prdx5 and Prdx6 have the most potent IL-23 inducing activities [76]. The IL-23 inducing activities of Prdx5 and Prdx6 are stronger than that of HMGB1, a well-known DAMP [76]. Because the cysteine residues in Prdxs are not required for their ability to act as DAMPs to induce IL-23 in BMDCs, the role of Prdxs as DAMPs seems to be independent of their antioxidant activity [76]. Rather, several specified regions (β4 sheet and α3 helix regions) located on the surface of Prdxs that are conserved among species, seem to be responsible for their ability to act as DAMPs [76]. A later study by Knoops et al. has confirmed the direct interaction between Prdx5 and TLR4 using the technique of atomic force microscopy in living macrophage cells [98]. 

### 2.6. Prdx6

It has recently been demonstrated that Prdx6 is involved in the suppression of ferroptosis, another form of inflammatory cell death [99]. Using the lentivirus-mediated inducible expression of siRNA for Prdx6, Lu et al. demonstrated that Prdx6 knockdown increases lipid hydroperoxide (LOOH) levels in H1299 human lung cancer cells treated with a ferroptosis inducer, erastin [99]. Prdx6 knockdown did not affect the basal levels of LOOH, suggesting that Prdx6 is not involved in the elimination of LOOH in healthy cells [99]. In contrast, the increase in LOOH in erastin-treated and Prdx6 knockdown cells was cancelled by ferrostatin-1, a ferroptosis inhibitor [99], suggesting ferroptosis in Prdx6 knockdown cells. Prdx6 can reduce LOOH through two mechanisms: as a peroxidase and as a phospholipase A_2_ (PLA_2_) [100,101,102]. Importantly, the increase in LOOH by Prdx6 knockdown was cancelled by iPLA_2_ (calcium-independent PLA_2_) inhibitor MJ-33, suggesting that the LOOH eliminating role of Prdx6 in ferroptosis is not dependent on its peroxidase activity, but rather on its iPLA_2_ activity [99]. Prdx6 has been shown to contribute to LPS-induced acute lung injury through its iPLA_2_ activity [103,104]. In contrast, it has also been reported that Prdx6 KO aggravates CLP-induced acute lung injury [105]. Thus, the true role of Prdx6 in pathogenesis needs to be clarified in the future, although its role seems to be highly context-dependent.

## 3. Conclusions

Sepsis, and other inflammatory disorders such as trauma and DILI, is regulated by Prdxs both intra- and extra-cellularly. Prdxs are involved in both propagation and regression of inflammation during sepsis. Based on the literature cited in this review, it seems possible to conclude that when Prdxs are localized within cells or vesicles, they generally suppress immune responses by suppressing inflammasome activation as well as generating mitochondrial ROS. On the other hand, Prdxs promote inflammation by acting as DAMPs when they are released from cells or vesicles. These unique two-sided features of Prdxs might make it difficult to determine whether Prdxs are beneficial or detrimental for the prevention of sepsis. More detailed elucidation of the regulatory mechanisms of the pathogenesis of sepsis and other inflammatory disorders should pave the way toward the use of Prdxs as a target of therapy for sepsis.

## Figures and Tables

**Figure 1 antioxidants-11-00126-f001:**
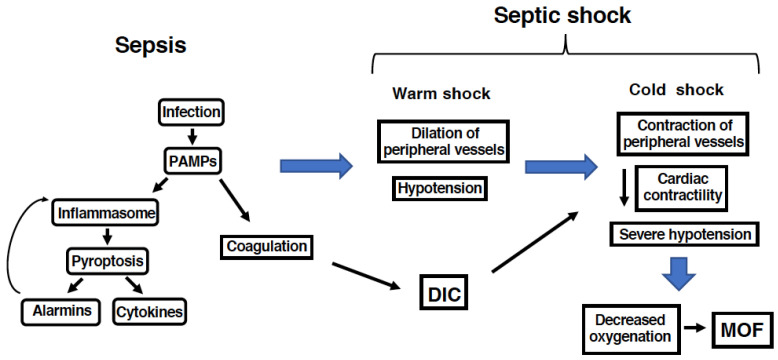
Pathophysiology of sepsis and septic shock. Sepsis is initiated by microbial infection. Typically, bacterial endotoxins, which are recognized by inflammatory cells as pathogen-associated molecular patterns (PAMPs), activate inflammasomes in the cells. Inflammasome activation leads to not only the release of cytokines as well as alarmins, but also to cell death through pyroptosis. The activation of coagulation is another characteristic of sepsis. Excessive and dysregulated inflammatory responses, as well as excessive coagulation, lead to the occurrence of many symptoms such as hypotension and disseminated intravascular coagulation (DIC). The early stage of septic shock (warm shock) is associated with low peripheral vascular resistance, while the later stage of septic shock (cold shock) is associated with increased systemic vascular resistance. These symptoms finally result in severe hypotension and life-threating multiple organ failures (MOF), resulting in the high mortality of this disease.

**Figure 2 antioxidants-11-00126-f002:**
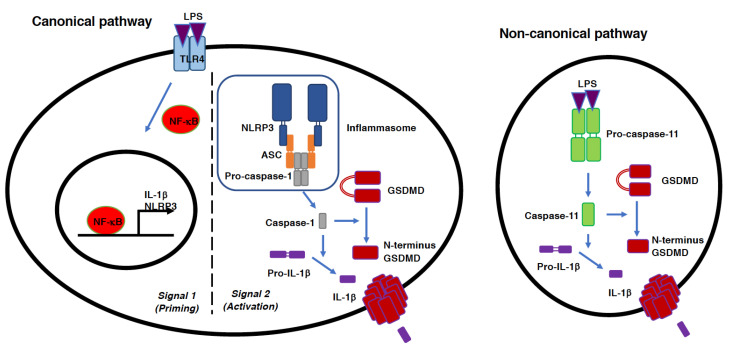
Canonical and non-canonical pathways of inflammasome activation and pyroptosis. The canonical inflammasome activation pathway is ordinally divided into two steps, the priming and activation steps. The priming step, also referred to as signal 1, involves the ligation of a PRR, such as TLR4, by PAMPs or DAMPs, such as LPS and HMGB1. PRR activation leads to the transcriptional induction of the genes for inflammatory cytokines (IL-1β and IL-18, etc.) and inflammasome components (NLRP3, ASC, and caspase-1). NF-κB is the main transcription factor mediating this induction. The activation step, also referred to as sign 2, consists of the assembly of the inflammasome and the resultant activation of capsase-1 from its pro-form (pro-caspase-1). Activated caspase-1 further activates IL-1β from its pro-form (pro- IL-1β). Caspase-1 also cleaves GSDMD to generate the N-terminus fragment of GSDMD. The proteolytic activation of GSDMD results in the oligomerization of the N-terminus fragment of GSDMD on the plasma membrane. The mature form of IL-1β is released from the cells through plasma membrane pores, which consist of GSDMD oligomers. The non-canonical pathway of inflammasome activation is triggered by the direct sensing of intracellular LPS by caspase-11. The sensing of LPS by caspase-11 leads to the activation of this caspase from its pro-form (pro-caspase-11). Activated caspase-11 can activate IL-1β as well as GSDMD in the same manner as the canonical pathway. Both canonical and non-canonical inflammasome activation finally leads to cell death, called pyroptosis, which is associated with plasma membrane rupture.

**Figure 3 antioxidants-11-00126-f003:**
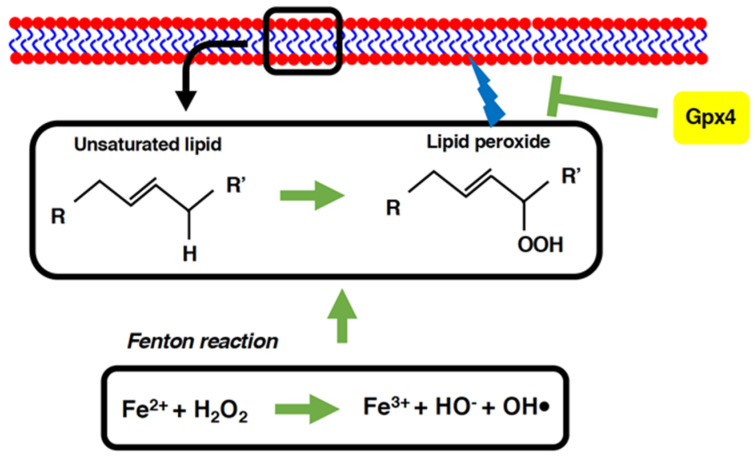
Lipid peroxidation leads to ferroptosis. The generation of reactive oxygen species (ROS) and the formation of lipid peroxides take place during ferroptosis. Because the ferrous iron (Fe^2+^) is required for ferroptosis, the generation of hydroxy radicals through the Fenton reaction is believed to be involved in ferroptosis. Usually, lipid peroxidation occurs in unsaturated lipid molecules, because the electrons in unsaturated lipids are relatively easy to abstract as compared to those in saturated lipids. Glutathione peroxidase 4 (Gpx4) suppresses ferroptosis by reducing the formation of lipid peroxides. The mechanism connecting lipid peroxidation and cell death (rupture of the plasma membrane) remains to be elucidated.

**Figure 4 antioxidants-11-00126-f004:**
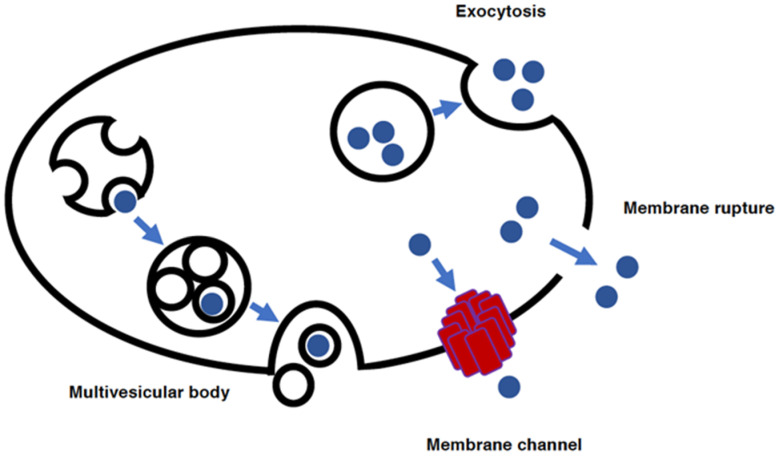
Mechanisms for the release of DAMPs from cells. DAMPs can be released by inflammasome-dependent mechanisms. Upon inflammasome activation and GSDMD pore formation in plasma membranes, DAMPs with relatively small molecular masses can be released through the pores. After the inflammasome activation proceeds and cell death occurs, all DAMPs can be released from the cells when the cell membrane ruptures. In addition to these inflammasome-based mechanisms, DAMPs can be released through exocytosis. The exocytosis of secretory vesicles is the ordinary process responsible for the secretion of intracellular molecules from cells. Multivesicular bodies are structures generated by the inward budding of the membrane of endosomes toward their luminal space. Vesicles generated within endosomes contain intracellular contents within their structures, and these vesicles can be released into the extracellular space through exocytosis.

**Figure 5 antioxidants-11-00126-f005:**
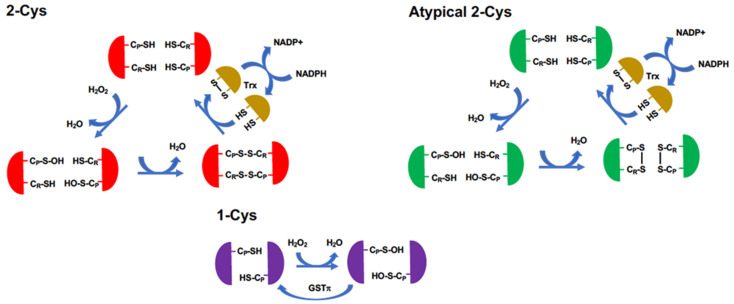
Mechanisms for the reduction of peroxides by Prdxs. The 2-Cys Prdxs, comprising Prdx1–4, are dimers in which the monomers are arranged in an anti-parallel fashion to one another. Each monomer possesses two distinct Cys residues: a peroxidatic Cys (C_P_) involved in the reduction of peroxides and a resolving Cys (C_R_) involved in the reduction of the oxidized C_p_ through the formation of an intermolecular disulfide bond between C_P_ and C_R_. Atypical 2-Cys Prdx (Prdx5) is distinct from the typical 2-Cys Prdxs in its manner of disulfide formation between C_P_ and C_R_; atypical 2-Cys Prdx forms an intramolecular disulfide bond during the reduction of C_P_ by C_R_. In the reduction processes of both typical and atypical 2-Cys Prdxs, Trx is involved in the reduction of the disulfide bond. The 1-Cys Prdx (Prdx6) has the unique characteristic of having only one Cys involved in the reduction of peroxides. Prdx6 utilizes GSTπ to reduce oxidized C_P_.

**Figure 6 antioxidants-11-00126-f006:**
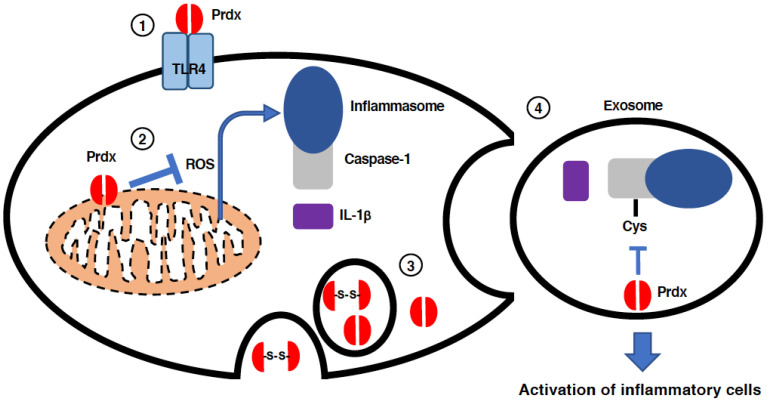
Intracellular and extracellular roles of Prdxs in the regulation of inflammatory responses. When released from cells into the extracellular space, Prdxs act as DAMPs by binding directly to TLRs including TLR4 (①). Prdx3, which resides in mitochondria, eliminates the ROS generated in damaged mitochondria thereby suppressing inflammasome activation (②). Intracellular Prdxs can be exocytosed from cells as disulfide bonded dimers (③). Prdx4 is released from inflammatory cells as a component of exosomes, in which functional inflammasomes are retained to activate other inflammatory cells (④).

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
