# Peer review of "The Role of Peroxiredoxins in the Regulation of Sepsis"

_antioxidants, 2022, doi:10.3390/antiox11010126_

Round 1

Reviewer 1 Report

Peroxiredoxins (Prdxs) are proteins that were first described as intracellular anti-oxidative enzymes involved in cellular redox homeostasis through the reduction of peroxides (hydrogen peroxide, lipid peroxides, or nitrogen peroxides).

Here, the data showing that Prdxs have both intracellular and extracellular functions, and anti- or pro-inflammatory roles is reviewed. It is particularly interesting the review of their role as molecules with Danger- (or Damage-) Associated Molecular Pattern (DAMP) properties that help in sterile or sepsis associated inflammation.

The review is well structured and the contents adequate, with a logic order. Its six figures are clear and illuminate some of the key points of the topics involved, from inflammation triggering to anti-oxidative mechanisms. The language is usually concise and clear, and the references are up to date and relevant.

Minor points.

There are some style redundancies in the text that should be corrected, i.e. lines 12-13: “Prdxs are also involved in the regulation of cellular signal transduction by modulating the activities of a panel of molecules involved in the signal transduction process.”

In line 265 of the text, the sentence “…..suggested the possibility that Prdx1 itself is not ligand of TLR4, consideration of but rather that some as yet unknown molecules associated with Prdx1 might be the true ligands [74].” should be corrected.

Author Response

Peroxiredoxins (Prdxs) are proteins that were first described as intracellular anti-oxidative enzymes involved in cellular redox homeostasis through the reduction of peroxides (hydrogen peroxide, lipid peroxides, or nitrogen peroxides).

Here, the data showing that Prdxs have both intracellular and extracellular functions, and anti- or pro-inflammatory roles is reviewed. It is particularly interesting the review of their role as molecules with Danger- (or Damage-) Associated Molecular Pattern (DAMP) properties that help in sterile or sepsis associated inflammation.

The review is well structured and the contents adequate, with a logic order. Its six figures are clear and illuminate some of the key points of the topics involved, from inflammation triggering to anti-oxidative mechanisms. The language is usually concise and clear, and the references are up to date and relevant.

(Reply)

We appreciate for your kind evaluation.

Minor points.

There are some style redundancies in the text that should be corrected, i.e. lines 12-13: “Prdxs are also involved in the regulation of cellular signal transduction by modulating the activities of a panel of molecules involved in the signal transduction process.”

(Reply)

Thank you for your advice. We rephrased this sentence to avoid style redundancies. Although we could not find anything else, we are willing to correct if there are still other redundancies.

In line 265 of the text, the sentence “…..suggested the possibility that Prdx1 itself is not ligand of TLR4, consideration of but rather that some as yet unknown molecules associated with Prdx1 might be the true ligands [74].” should be corrected.

(Reply)

We corrected the sentence.

Reviewer 2 Report

Summary: This manuscript is a review article describing Peroxiredoxins and their role in sepsis and other inflammatory disorders. The manuscript gives an overview of sepsis and the inflammatory response.

Critique: This is well written review describing Peroxiredoxins and the known inflammatory response of sepsis. Regarding the introduction, I would recommend adding information on the treatment of sepsis and the role of the sepsis bundle that is now generally accepted. The strength of this review is the multi-faceted role of Peroxiredoxins of damping the inflammatory response intracellular but becoming pro-inflammatory in the extracellular compartment. Since exosomes are being implicated in the pathogenesis of sepsis, is there any references describing the role of Peroxiredoxins and exosomes? Do exosomes carry Peroxiredoxins? Initially, I thought more information could be included on the role of Peroxiredoxins in sepsis, but a Pub med search reveals the knowledge is limited, so I thought the authors did a good job in describing what is current. The number and quality of the references appear to be complete.

The only change I would consider is the title; the authors include “other Inflammatory disorders” but I am unclear what they include. Sepsis is the major focus, they mention intracerebral hemorrhage and a few sentences on liver damage, but not much else, so I would consider re-wording the title to be more specific.

Author Response

Summary: This manuscript is a review article describing Peroxiredoxins and their role in sepsis and other inflammatory disorders. The manuscript gives an overview of sepsis and the inflammatory response.

Critique: This is well written review describing Peroxiredoxins and the known inflammatory response of sepsis.

(Reply)

We appreciate very much for your evaluation.

Regarding the introduction, I would recommend adding information on the treatment of sepsis and the role of the sepsis bundle that is now generally accepted.

(Reply)

Many thanks for your feedback. Since the purpose of this review article is to provide the information about pathophysiological mechanism of sepsis, we prefer only making mention to the sepsis bundle with appropriate references (lines 48 and 49).

The strength of this review is the multi-faceted role of Peroxiredoxins of damping the inflammatory response intracellular but becoming pro-inflammatory in the extracellular compartment. Since exosomes are being implicated in the pathogenesis of sepsis, is there any references describing the role of Peroxiredoxins and exosomes? Do exosomes carry Peroxiredoxins? Initially, I thought more information could be included on the role of Peroxiredoxins in sepsis, but a Pub med search reveals the knowledge is limited, so I thought the authors did a good job in describing what is current. The number and quality of the references appear to be complete.

(Reply)

Thank you for thoughtful discussion. We agree to your opinion that the knowledge of relevant issue is currently limited.

The only change I would consider is the title; the authors include “other Inflammatory disorders” but I am unclear what they include. Sepsis is the major focus, they mention intracerebral hemorrhage and a few sentences on liver damage, but not much else, so I would consider re-wording the title to be more specific.

(Reply)

Thanks for your advice. We deleted “other Inflammatory disorders” from the title.
